# Diffuse Idiopathic Skeletal Hyperostosis (DISH): Role of Logopedic Rehabilitation in Dysphagia

**DOI:** 10.3390/jpm13060994

**Published:** 2023-06-13

**Authors:** Massimo Mesolella, Sarah Buono, Roberto D’Aniello, Annarita Ascione, Gaetano Motta, Salvatore Allosso

**Affiliations:** 1Unit of Otorhinolaryngology, Department of Neuroscience, Reproductive Sciences and Dentistry, University Federico II of Naples, 80131 Naples, Italy; sarah.buono1990@gmail.com (S.B.); robedan88@gmail.com (R.D.); ascioneannarita90@gmail.com (A.A.); 2Unit of Otorhinolaryngology, University Luigi Vanvitelli, 80131 Naples, Italy; gaetano.motta@unicampania.it

**Keywords:** Forestier’s Syndrome, DISH (diffuse idiopathic skeletal hyperostosis), dysphagia, logopedic and postural rehabilitative treatment

## Abstract

Forestier’s disease is a systemic, degenerative metabolic condition, which is still insufficiently investigated and understood, characterized by the progressive ossification of the ligaments and entheses. This case describes a 63-year-old man admitted to our department after several years of failed attempts to obtain a definitive diagnosis presenting with a painless mass in the pre-auricular region, gradually worsening dysphonia, severe dysphagia relating to solids, stiffness, and mild pain at the back of the neck. After performing further diagnostic tests, in addition to a pleomorphic adenoma, the simultaneous presence of diffuse spondylarthrosis on the cervical spine was highlighted, with beak-like osteophytes at C2–C5 resulting in esophageal compression. Given the absence of upper digestive endoscopy abnormalities, we initiated an intense logopedic and postural rehabilitative treatment, which led to a significant improvement in the patient’s dysphagia symptoms. In addition, we limited the use of medical therapy to solely indomethacin in order to control the osteophytic process.

## 1. Introduction

Forestier’s Syndrome, known under the acronym DISH (diffuse idiopathic skeletal hyperostosis), is a systemic degenerative pathology affecting the entheses characterized by the altered ossification of the insertion points of ligaments, tendons, and joint capsules. Through autopsies, this pathology has been found in 18% of the Western population; it increases in frequency with age and mainly affects men (2:1). It mainly affects the thoracic spine, followed by the cervical region in 75% of cases.

Since DISH is asymptomatic in most people, its prevalence in the general population is surely underestimated. However, since it is often associated with metabolic alterations and cardiovascular diseases, it can be a major indicator of metabolic alterations that are dangerous for the patient [1,2,3,4].

Several metabolic factors are involved in DISH: central obesity, type II diabetes, hypertension, and metabolic syndrome. These are morbid conditions that can lead to an increase in pro-inflammatory cytokines responsible for amplifying metabolic problems. An important role is played by central obesity; it is known that abdominal fat is a determinative agent in hypertension and insulin resistance [3].

Since macroscopic examinations of human remains have shown traces of DISH in people of high social rank and even in the Pharaoh Ramses II and in Neanderthals, it is probable that there is strong evidence linking the development of this disease with risk factors such as obesity, diabetes, and hypertension [5]. This phenomenon would also explain the higher incidence in Western countries [1].

DISH previously has previously been described in the literature under different names:-Spondilitis ossificant ligamentosa: When studying 282 patients with ossification of the vertebral ligaments, Oppenheimer identified that the calcification or ossification mainly involved the anterior longitudinal ligament in the thoracic region, while the lumbar region was involved to a lesser extent and near absence was noted in the cervical region. His studies showed that the ossification of the ligament occurred as a consequence of vertebral immobility. For this reason, immobilization favored the transformation of the connective tissue of the ligament into bone tissue. Therefore, it was surmised that spondylitis ossificans was not an isolated phenomenon but a condition secondary to immobility [6].-Physiologic vertebral ligamentous calcification: Smith, studying 53 patients over the age of 60 with pain and stiffness of the spine, realized from radiological images that these patients showed ossification of the anterior longitudinal ligament and osteophytosis, both of which were in the absence of osteoporosis. The ossified ligament appeared “flaccid” and distinct from the anterior margins of the adjacent vertebrae. Smith also noted that the reduction in spinal column movement in elderly patients could be the cause of ligament degeneration [7].-Generalized juxta-articular ossification of vertebral ligaments (senile) ankylosing hyperostosis of the spine: Forestier and colleagues introduced the term senile from the study of 200 elderly patients, 65% of whom were male [8].

Since vertebral ankylosis is evident only through radiography and because extraspinal changes appear even without vertebral abnormalities, the term diffuse idiopathic skeletal hyperostosis (DISH) has been introduced [9].

The diagnosis of this disease requires the presence of radiological abnormalities (observed via Rx, CT, and/or MRI); its etiopathogenesis, which is still unknown, is probably chronic micro-traumatism associated with the most common metabolic alterations of adulthood (altered metabolism of vitamin A and the release of GH, IGF-1, and DMII). The final event that involves all the units mentioned above is always the generation of new bone tissue [1].

When present, the general symptoms are mild in nature and include back pain and axial stiffness [10]. In the highest-grade forms, the clinical otolaryngologic manifestations can alter aerodigestive functions, leading to dysphonia, severe dyspnea, or dysphagia, thereby compromising the normal mobility of the musculoskeletal system of the cervicofacial district. Most patients are asymptomatic, are capable of mild spinal movement, or possess joint pain. The disease has long been thought to be an asymptomatic condition, which is usually detected incidentally during old age [10,11].

The purpose of our article is to verify the validity of a conservative treatment, in our case, speech therapy, for the improvement of dysphagia in a patient affected by DISH.

## 2. Case Report

A 63-year-old male patient who had been a smoker of 10 cigarettes a day for twenty years and was hypertensive (under pharmacological treatment), hypercholsterolemic, overweight, and suffering from type 2 diabetes mellitus presented to our department complaining of progressively worsening dysphonia, severe dysphagia mainly with respect to solids with which he had been afflicted for the last year, aspiration, globus sensation, cervical pain, and a slowly progressing asymptomatic swelling in the pre-auricular region.

He also complained of mild pain at the back of the neck, which was associated with scapulohumeral impairment. Evaluation through flexible (fiber-optic) laryngoscopy revealed the presence of a mass protruding from the posterior wall of the rhinopharynx causing a critical narrowing of the oropharyngeal lumen and the laryngeal aditus. Mucosa appeared normal with no other abnormalities detected (Figure 1).

We performed a lateral–cervical spinal X-ray, a CT scan (Figure 2), and MRI (Figure 3) of the head, neck, and chest, which demonstrated anterior osteophytosis resulting in the compression of the esophagus between levels C2 and C5 and widespread axial spondylarthrosis. We also performed an esophago-gastric endoscopy, which showed a normal and undamaged esophagus, with whitish-pink cardiac mucosa and no motility disorders (Figure 4); in addition, the lumen appeared to present an ab extrinsic compression at its proximal tract. Given the absence of upper digestive endoscopy abnormalities, we initiated a logopedic and postural rehabilitative treatment.

Due to the presentation of dysphagia, we provided the questionnaire EAT-10 to the patient for the subjective evaluation of the disorder. The EAT-10 test is a standard test for monitoring a patient’s level of swallowing [12]. The test is composed of 10 items; for each one, the patient provides an answer corresponding to scores ranging from 0 to 4. A score of 0 means there are problems are present, while 4 means that the patient has severe injuries [12] (Appendix A).

The highest score for each submission is 40. If the final score is equal to or greater than three, the patient could have problems swallowing safely.

The patient reported in this study answered the items just mentioned: for items 1, 2, 4, 6, and 7, the answer was 2 points. For all the other items, the answer was 1 point on the scale (a total of 15 points)

The patient was submitted to logopedic therapy twice a week a total of 30 times in four months. During the treatment, the patient underwent tongue praxis to magnify the movement of retropulsion. In order to ensure that the patient could swallow safely, he was required to assume a posture consisting of an upright trunk with his neck turned to the left side (essentially a supraglottic maneuver).

After rehabilitation, the patient related a significant improvement in dysphagia symptoms and aspiration. We limited the use of medical therapy to indomethacin alone in order to control the osteophytic process.

The EAT-10 test was provided to the patient one more time after the treatment (after four months), and its answers corresponded to an average point of one most of the time and zero for the third and eighth times (amounting to a total of eight points).

## 3. Discussion

DISH is considered a rheumatic disorder. Although its pathogenesis is not yet fully understood, its seems to share an inflammatory pathway with metabolic syndrome; in fact, diabetes, hypercholesterolemia, and hyperuricemia appear to be risk factors [3,10].

Diffuse idiopathic skeletal hyperostosis, which affects men more frequently than women [9,13], is a condition characterized by the ossification of the enthesis (enthesopathy) that can cause a variety of symptoms, such as dysphagia, as a result of cervical osteophytes compression. In fact, protruding osteophytes have been demonstrated to precipitate disorders that may lead to the dysfunction of the swallowing mechanism [11,14,15,16,17]. Resnick reported a 28% prevalence of dysphagia related to cervical spine involvement in patients with DISH; only 4% of these patients had dysphagia as their presenting complaint [9]. Osteophyte-induced dysphagia associated with DISH or spondylosis may occur through several different pathogenic mechanisms, such as the luminal impingement of the esophagus by a large osteophyte, periesophageal inflammation and edema caused by pharyngoesophageal irritation via osteophytes, a small osteophyte that is specifically located at a fixed position with respect to the esophagus, and pain and muscle spasms due to irritation by an osteophyte, which may cause narrowing [18]. Considering the importance of differential diagnosis, especially among elderly patients, it is critical not to overlook the possibility of a malignant etiology [19,20]. In fact, among the various conditions that are accounted for in a differential diagnosis, in addition to abscesses of the retropharyngeal space and a possible Zenker’s diverticulum, it is important to consider oropharyngeal and cervical esophageal tumors [1].

There are no specific laboratory investigations for this disease; only radiological studies can aid its diagnosis. Indeed, the diagnostic criteria are ossification within the anterior longitudinal ligaments of at least four continuous vertebral bodies with the preservation of disc space height and the absence of degenerative changes. Despite radiographic evidence, the management of DISH is closely related to the degree of symptoms. This can be based on conservative treatment such as dietary modifications, swallowing therapy, and the prescription of NSAI drugs; alternatively, in the case of failure, a surgical treatment is preferred [10].

The most commonly used surgical procedure to address dysphagia is osteophytectomy. Although it is a well-tolerated procedure, about 4% of patients who undergo surgery present with dysphagia again as a post-operative complication. The mechanisms may be due to either post-surgical edema and consequent scarring or to spasms of the esophageal muscles following surgery. It is possible to overcome this muscular issue by performing a simultaneous cricopharyngeal myotomy [21]. However, surgical treatment is a valid alternative after the failure of the conservative management of a patient, especially when dysphagia worsens [22].

After review by a multidisciplinary board (which included an otolaryngologist, a radiologist, an internal medicine doctor, and a neurosurgeon), we defined our approach to be conservative. We started a multiple course of indomethacin in order to slow down bone spur growth alongside an intensive treatment of logopedic, postural rehabilitation (head posture control) to reduce the dysphagia-related symptomatology. For individuals with diffuse idiopathic skeletal hyperostosis with dysphagia, rehabilitative logopedic techniques are compensatory [23,24] and include head-positioning exercises, strengthening the tone of the tongue and muscles engaged in swallowing, and changes in the consistency of the bolus.

## 4. Conclusions

DISH is an uncommon cause of dysphagia with a male preponderance that mostly affects older individuals [25,26,27,28]. In daily practice, the corresponding treatment approach is based on knowledge gathered from other disease therapies and empirical approaches applied to single patients, and it is strictly based on the severity of the presented symptoms. In our experience, a multidisciplinary approach has been found to be the only way to manage the disease. The role of the otolaryngologist, especially in differential diagnosis and, subsequently, during the follow-up, has facilitated both the diagnosis and the choice of the most fitting therapeutic conduct, reducing the repetition of unnecessary examinations for the patient and the short and long control termination of therapy, offering excellent results.

We find it interesting that a multidisciplinary approach together with an intense logopedic rehabilitation can lead to profound and longstanding improvements of dysphagia, without the need to proceed with any surgical treatment (which could be a key determinant, especially among elderly patients that are often exposed to surgical risks).

## Figures and Tables

**Figure 1 jpm-13-00994-f001:**
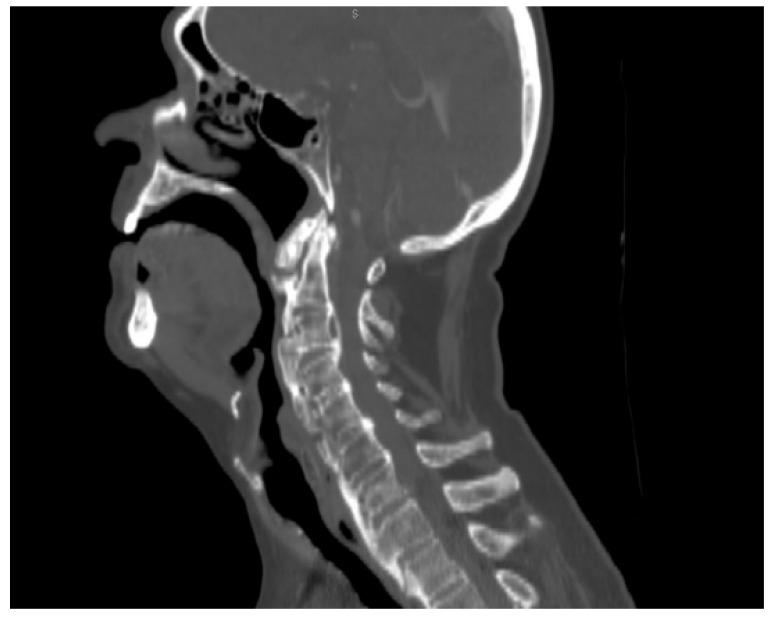
Sagittal CT: hyperostotic anterior longitudinal spinal ligament with large osteophytes compressing the aerodigestive tract. The lesion extended from C2–C5.

**Figure 2 jpm-13-00994-f002:**
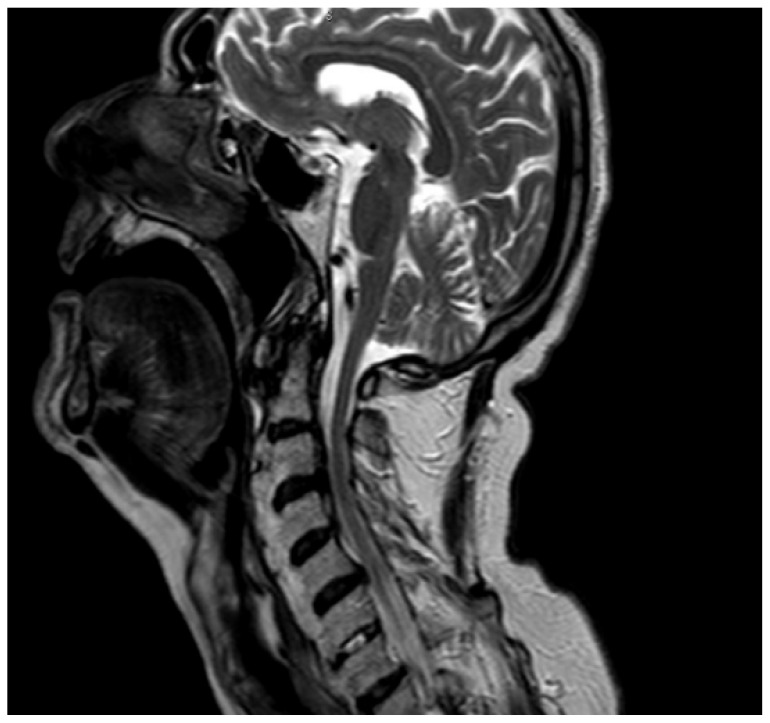
MRI showing diffuse spondyloarthrosis and osteophytes corresponding to the C2–C5 tract with a narrowing of the pharyngolaryngeal lumen.

**Figure 3 jpm-13-00994-f003:**
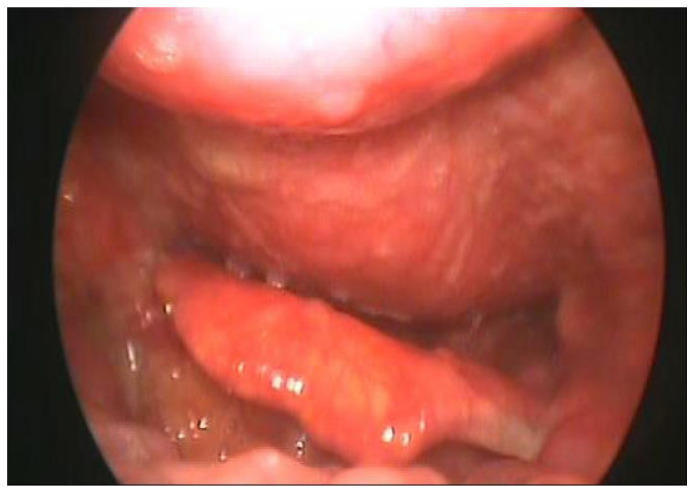
A bulging posterior pharyngeal wall contacting the base of the tongue and the epiglottis.

**Figure 4 jpm-13-00994-f004:**
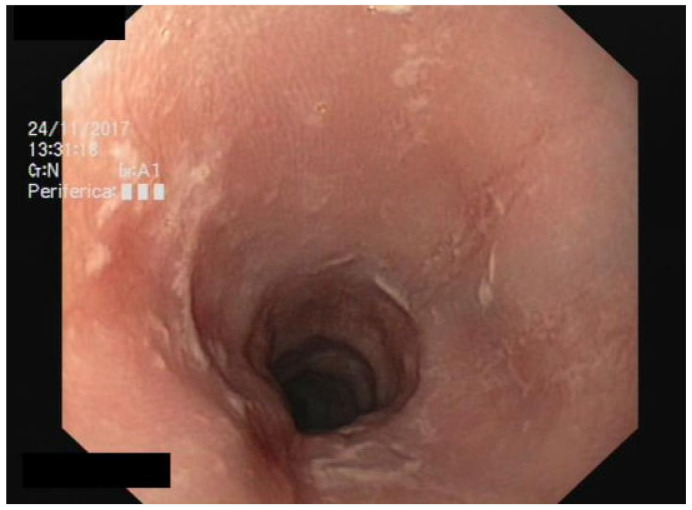
Esophago-gastric endoscopy showing normal and undamaged mucosa of the esophagus.

## Data Availability

Data sharing not applicable. No new data were created or analyzed in this study. Data sharing is not applicable to this article.

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
