# Peer review of "Diffuse Idiopathic Skeletal Hyperostosis (DISH): Role of Logopedic Rehabilitation in Dysphagia"

_jpm, 2023, doi:10.3390/jpm13060994_

Round 1
Reviewer 1 Report
In this case report the authors report an interesting case of dysphagia in a patient with diffuse idiopathic skeletal hyperostosis (DISH) who was treated with conservative treatment, which included speech therapy and postural changes. While many cases of dysphagia have been reported to date, this case adds to the ever growing literature of dysphagia in DISH. I have a few remarks on the manuscript, with regards to the effect of speech and logotherapy on the resolvement of dysphagia.
In the abstract, introduction, and discussion the authors mention that DISH is a non-inflammatory disease. While it has been theorized for decades that the pathophysiology is non-inflammatory, recent evidence is emerging highlighting the possibility of an inflammatory component in its pathogenesis (PMID: 33496875), which has been supported by the increased visceral adiposity markers (PMID: 34357130) and inflammatory markers in DISH (PMID: 36933072). Therefore, I would suggest to formulate the text more nuanced and remove the non-inflammatory aspect of the disease.
Please include the comorbidities of the patient in the case report
How did the symptoms of dysphonia and pain improve or worsen after treatment?
It is unclear in the text at which time points the patient completed the EAT-10 questionnaire. If the EAT-10 questionnaire was only measured at one time point, and not before and after conservative treatment, it would be better to exclude this out of the case report.
It should be mentioned in the discussion section that the improvement of dysphagia is most likely the result of the postural changes associated with conservative management. Speech therapy and logopedic treatment would most likely influence the symptoms of dysphonia (see also my comment above on the dysphonia symptoms after treatment), but not the swallowing disorder to solids.
In line with the comment above, this should also be removed from the conclusion.
The discussion section mentions specifically that the patient was treated with a physiatrist. This however, is not mentioned in the results section of the manuscript.
Please also add that patients receive surgical treatment after conservative management has failed, when symptoms of dysphagia progress or neurological symptoms such as myelopathy are found (PMID: 35283294).
It should be mentioned in the discussion that a cricopharyngeal myotomy is not standardly performed, and usually only reported in up to 9 cases of 419 (2%). Recent literature has shown that dysphagia recurrence is lower than 10% (at 4%), and it is not always associated with the regrowth of osteophytes or bone spurs at the level of the spine (PMID: 35283294).
The discussion mentions the role of a multidiscplinary team in the management of DISH. Please mention which specialties are included in the process, e.g. radiology, orthopedic or neurological surgery etc
A few grammatical and spelling errors encountered in the text, which require revision.
Author Response
In this case report the authors report an interesting case of dysphagia in a patient with diffuse idiopathic skeletal hyperostosis (DISH) who was treated with conservative treatment, which included speech therapy and postural changes. While many cases of dysphagia have been reported to date, this case adds to the ever growing literature of dysphagia in DISH. I have a few remarks on the manuscript, with regards to the effect of speech and logotherapy on the resolvement of dysphagia.
In the abstract, introduction, and discussion the authors mention that DISH is a noninflammatory disease. While it has been theorized for decades that the pathophysiology is non-inflammatory, recent evidence is emerging highlighting the possibility of an inflammatory component in its pathogenesis (PMID: 33496875), which has been supported by the increased visceral adiposity markers (PMID: 34357130) and inflammatory markers in DISH (PMID: 36933072). Therefore, I would suggest to formulate the text more nuanced and remove the non-inflammatory aspect of the disease.
- I've made the requested changes and quoted the suggested articles.
Please include the comorbidities of the patient in the case report.
- I’ve made the requested changes. LINE 84
How did the symptoms of dysphonia and pain improve or worsen after treatment?
- LINE 122 : The EAT – 10 test was subjected to the patient one more time after the treatment (after four months), and its answers reported an average point of 1 in most of the times, and into 0 for the times number 3 and 8. (Total 8 points)
- We only evaluated the improvement in dysphagia on the basis of the EAT 10 questionnaire and the improvements were demonstrated by the lower score obtained (8 points). Improvement in hoarseness was not evaluated.
It is unclear in the text at which time points the patient completed the EAT-10 questionnaire. If the EAT-10 questionnaire was only measured at one time point, and not before and after conservative treatment, it would be better to exclude this out of the case report.
- The questionnaire was submitted before treatment and after treatment, i.e. 4 months after speech therapy rehabilitation. This point is well specified in the text. Removing this parameter does not make dysphagia symptoms objectionable.
It should be mentioned in the discussion section that the improvement of dysphagia is most likely the result of the postural changes associated with conservative management. Speech therapy and logopedic treatment would most likely influence the symptoms of dysphonia (see also my comment above on the dysphonia symptoms after treatment), but not the swallowing disorder to solids. In line with the comment above, this should also be removed from the conclusion.
- During the treatment, the patient was involved into tongue praxis to magnify the movement of retropulsion.
- It is not only the posture learned by the patient responsible for the improvement in swallowing but also the retropulsion exercises of the tongue (as explained). these mechanisms help to determine a supraglottic compensation for swallowing.
The discussion section mentions specifically that the patient was treated with a physiatrist. This however, is not mentioned in the results section of the manuscript.
- It was a transcription error. We wanted to understand that the patient, under the guidance of the speech therapist, learned both swallowing and head posture exercises.
Please also add that patients receive surgical treatment after conservative management has failed, when symptoms of dysphagia progress or neurological symptoms such as myelopathy are found (PMID: 35283294).
- Added in LINE 160
It should be mentioned in the discussion that a cricopharyngeal myotomy is not standardly performed, and usually only reported in up to 9 cases of 419 (2%). Recent literature has shown that dysphagia recurrence is lower than 10% (at 4%), and it is not always associated with the regrowth of osteophytes or bone spurs at the level of the spine (PMID: 35283294).
- I HAVE CORRECT THE PERCENTAGE.
The discussion mentions the role of a multidiscplinary team in the management of DISH. Please mention which specialties are included in the process, e.g. radiology, orthopedic or neurological surgery etc
- LINE 164

Reviewer 2 Report
In this article, the authors reported a clinical case of diffuse idiopathic skeletal hyperostosis (DISH) that is treated with logopedic and postural rehabilitation. I think that its theme is interesting; however, there are some concerns which have to be resolved before considering publication.
1. [Introduction and Conclusions] Introduction and conclusions in this article are too redundant and confusing. Authors should describe them simpler and more concise.
2. [Introduction] There is not the aim of the article in introduction.
3. [Case Report] Because it is a case report, author should provide more clinical information. Especially, authors ought to describe a medical history of the patient since they mentioned risk factors of DISH.
4. [Case Report] Authors described that dysphasia was improved only based on the EAT score though it is subjective evaluation. I think the findings of objective examination —for example, preoperative and postoperative videofluoroscopic examination— are needed. In addition, when was the EAT after rehabilitation evaluated? I wonder whether the effect of the rehabilitation continues after finishing it or not.
5. [Discussion] Authors negatively declared about surgical treatment for DISH with dysphasia. It is true that anterior cervical approach has some crucial risks; however, I think surgical treatment is occasionally needed in some cases. I recommend that authors should describe pros and cons of surgical and conservative treatments in detail.
6. [Overall] There are some typo.
Author Response
In this article, the authors reported a clinical case of diffuse idiopathic skeletal hyperostosis (DISH) that is treated with logopedic and postural rehabilitation. I think that its theme is interesting; however, there are some concerns which have to be resolved before considering publication.
- [Introduction and Conclusions] Introduction and conclusions in this article are too redundant and confusing. Authors should describe them simpler and more concise.
- have been corrected
- [Introduction] There is not the aim of the article in introduction.
- have been corrected LINE 83
- [Case Report] Because it is a case report, author should provide more clinical information. Especially, authors ought to describe a medical history of the patient since they mentioned risk factors of DISH.
- I HAVE ADDED line 86
- [Case Report] Authors described that dysphasia was improved only based on the EAT score though it is subjective evaluation. I think the findings of objective examination —for example, preoperative and postoperative videofluoroscopic examination— are needed. In addition, when was the EAT after rehabilitation evaluated? I wonder whether the effect of the rehabilitation continues after finishing it or not.
- Unfortunately the patient refused to undergo videofluoroscopy for this reason, our only treated case was evaluated only with the EAT-10 questionnaire. As explained in the article, a questionnaire was administered before the start of treatment and at the end of the same (after 4 months LINE 124). The results are still good today. The patient should repeat the oral examination in a few months and in that case the questionnaire will be administered again.
- [Discussion] Authors negatively declared about surgical treatment for DISH with dysphasia. It is true that anterior cervical approach has some crucial risks; however, I think surgical treatment is occasionally needed in some cases. I recommend that authors should describe pros and cons of surgical and conservative treatments in detail.
- OK. we think that medical treatment is a valid alternative especially in patients with various comforts or who refuse surgery.
- [Overall] There are some typo.
- I have correct

Round 2
Reviewer 2 Report
The authors mostly responded to my comment; however, they partially didn't. Especially, they ignored the comment 1. For example, the introduction has too much unnecessary information such as a term variation of DISH.
Although the other contents improved, the introduction and conclusion should be drastically revised.
Author Response
The authors mostly responded to my comment; however, they partially didn't. Especially, they ignored the comment 1. For example, the introduction has too much unnecessary information such as a term variation of DISH.
Although the other contents improved, the introduction and conclusion should be drastically revised.
Dear Reviewer,
I appreciate your comments and I think I was quite accurate already during the first lap.
For us, including the history and evolution of the diagnosis of DISH over time in the introductory paragraph is important for understanding the evolution of diagnostics and how the new knowledge acquired over time has changed the approach to the disease both in terms of diagnosis and therapy. This is why we have not removed this story in the introductory part because the natural history of the disease is responsible for the name of the disease and its evolution.
We don't understand how the conclusions can be "confusing" since they are concise and above all aim to describe the objective of our study. In other words, we have reiterated that DISH is a pathology that is not only of ENT interest but that can manifest itself with symptoms of the ENT sphere. For this reason, when a symptom such as dysphagia occurs, a treatment is necessary which, in our experience, is not said to be surgical. This is especially valid in the elderly patient with various comorbidities who may not benefit from surgery that is often successful but is not free from side effects such as worsening of dysphagia. Having had good results from speech therapy rehabilitation, we have described and demonstrated the non-surgical therapeutic alternative and above all the maintenance of the result over time.
I hope these clarifications can be useful for the consideration of the work.
